# FedOSAA: Improving Federated Learning with One-Step Anderson Acceleration[*]

Xue Feng[1], M. Paul Laiu[2], Thomas Strohmer[1]

[1] University of California, Davis, [2] Oak Ridge National Laboratory

xffeng@ucdavis.edu, laiump@ornl.gov, strohmer@math.ucdavis.edu

Federated learning (FL) is a distributed machine learning approach that enables multiple local clients and a central server to collaboratively train a model while keeping the data on their own devices. First-order methods, particularly those incorporating variance reduction techniques, are the most widely used FL algorithms due to their simple implementation and stable performance. However, these methods tend to be slow and require a large number of communication rounds to reach the global minimizer. We propose FedOSAA, a novel approach that preserves the simplicity of first-order methods while achieving the rapid convergence typically associated with second-order methods. Our approach applies one Anderson acceleration (AA) step following classical local updates based on first-order methods with variance reduction, such as FedSVRG and SCAFFOLD, during local training. This AA step is able to leverage curvature information from the history points and gives a new update that approximates the Newton-GMRES direction, thereby significantly improving the convergence. We establish a local linear convergence rate to the global minimizer of FedOSAA for smooth and strongly convex loss functions. Numerical comparisons show that FedOSAA substantially improves the communication and computation efficiency of the original first-order methods, achieving performance comparable to second-order methods like GIANT.

## 1. Introduction

We consider a standard federated learning (FL) [1] architecture where a set of clients collaboratively work with a central server to train a model, with the objective function defined as follows:

$$\min_{\boldsymbol{w}\in\mathbb{R}^d} f(\boldsymbol{w}) = \sum_{k=1}^{K} \frac{N_k}{N} f_k(\boldsymbol{w}). \tag{1}$$

Here, $K$ is the number of clients, $N = \sum_{k=1}^{K} N_k$, and $f : \mathbb{R}^d \to \mathbb{R}$ represents the global function and is a weighted average of local clients' objective functions $f_k$. In various machine learning problems, such as classification and regression, $f_k$ could be the empirical risk minimization function which depends on the private data of each client, with $N_k$ representing the number of local data points. The minimum of $f$ is denoted as $f^*$.

Federated learning typically involves two phases that alternate periodically during training: (1) at iterate $\boldsymbol{w}^t$, clients perform local updates to minimize its own local loss function, and (2) the server aggregates these local updates $\boldsymbol{w}_k^t$ to update the global model as $\boldsymbol{w}^{t+1} = \sum_{k=1}^{K} \frac{N_k}{N} \boldsymbol{w}_k^t$, which is then transmitted back to the clients. This setup inherently presents two key challenges: minimizing communication overhead between clients and the server, and ensuring efficient local computation on each client's side. Addressing these challenges is critical for FL algorithms.

[*]This manuscript has been authored, in part, by UT-Battelle, LLC under Contract No. DE-AC05-00OR22725 with the U.S. Department of Energy. The United States Government retains and the publisher, by accepting the article for publication, acknowledges that the United States Government retains a non-exclusive, paid-up, irrevocable, world-wide license to publish or reproduce the published form of this manuscript, or allow others to do so, for United States Government purposes. The Department of Energy will provide public access to these results of federally sponsored research in accordance with the DOE Public Access Plan(http://energy.gov/downloads/doe-public-access-plan).

Various FL optimization algorithms have been proposed. The efficiency of first-order methods makes them particularly appealing, leading to many variants. The basic benchmark algorithm is FedAvg [1], where the client optimization phase involves multiple local (stochastic) gradient descent (GD) steps. However, FedAvg suffers from the client-drift effect, where the local iterates of each client tend to drift towards the minimum of their local loss function rather than the true global minimum, due to the heterogeneity between the local function $f_i$ and the global function $f$. To mitigate this issue and ensure convergence to the global minimum, FedAvg often requires a small or diminishing local stepsize [2, 3]. The optimal convergence rate to the global minimum for FedAvg is sublinear even for strongly convex problems.

An alternative approach is to apply variance reduction techniques to correct client drift in local updates. Examples include SAG [4], SAGA [5], FedSVRG [6] (equivalently, FedLin [7]), SCAFFOLD [2], and FedProx [8], etc. These methods typically exhibit better convergence performance and are less affected by heterogeneity. For instance, the convergence of FedSVRG improves to a linear rate with a constant local stepsize under arbitrary heterogeneity. Despite these improvements, first-order methods generally suffer from slow convergence even with fine-tuned local stepsizes, causing a high number of communication rounds needed to reach the global minimizer.

This motivates the development of acceleration methods such as momentum-based methods [9, 10] and second-order methods. The latter category includes algorithms such as Disco [11], DANE [12], GIANT [13], FedNL [14], and the Federated Quasi-Newton method [15]. These Newton-type methods achieve faster convergence by utilizing curvature information, often in combination with variance reduction techniques to find the global minimizer [13, 15]. For instance, during local training, GIANT solves a local Newton direction with the global gradient transmitted from the server instead of using the local gradient. However, despite their advantages, second-order methods may face challenges such as accessing the Hessian.

In this work, we are interested in developing a method that is easy to implement and enjoys fast convergence. Our motivation is from Anderson acceleration (AA), which is also known as Anderson mixing and is typically used to accelerate conventional fixed-point iterations by mixing history points. It can be regarded as a quasi-Newton method whose approximate Jacobian inverse satisfies a multisecant equation [16]. AA is known for its simplicity and significantly improved convergence performance, with wide applications in fields such as electronic structure calculations [16, 17], fluid dynamics [18], geometrical optimization [19], and more recently machine learning [20, 21]. We note that a recent study on the Alternating Anderson-Picard (AAP) method [22] demonstrated that AAP effectively approximates the Newton-GMRES directions at each step when the residual of the fixed-point iteration is small. This highlights the potential of AAP as an alternative to Newton-type methods, as it does not require access to the Hessian. These qualities make Anderson acceleration (AA) particularly attractive for machine learning applications. However, the use of AA in distributed optimization problems, including federated learning, has been relatively limited thus far.

Our contributions in this paper can be summarized below:

- We proposed a novel method called FedOSAA, which applies one Anderson acceleration (AA) step following classical local updates based on first-order methods with variance reduction such as FedSVRG and SCAFFOLD, during local training. This AA step is able to leverage curvature information from the history points and gives a new update that approximates the Newton-GMRES direction, thereby significantly accelerating the convergence. We note that the Newton-GMRES direction here is equivalent to the Newton-MINRES direction since the Hessian matrix is symmetric [23].

- We established a local linear convergence rate of FedOSAA to the global minimizer for smooth and strongly convex loss functions.

- We provide numerical examples demonstrating that FedOSAA improves the communication and computation efficiency of original first-order methods, achieving performance comparable to second-order methods like GIANT.

## 2. FedOSAA: Related Work and Algorithm

In this section, we outline the motivation behind our proposed algorithm. We begin by reviewing classical first-order federated learning (FL) methods that leverage variance reduction techniques to mitigate client drift. These methods perform (stochastic) gradient descent on a "corrected" local objective function, enabling accelerated convergence to the global minimizer even with aggressive local updates. However, as the number of local epochs increases, the computational cost for each client grows. Next, by reformulating the gradient descent step as a fixed-point iteration, it is natural to apply AA to speed up the local training by mixing the history points generated by the (stochastic) gradient descent steps. Finally, we introduce the proposed FedOSAA algorithm, which integrates an additional AA step into existing first-order methods to improve the convergence.

### 2.1. Variance-Reduced First-Order Methods

In the centralized setting, the ideal update is $\boldsymbol{w}^{t+1} = \boldsymbol{w}^t - \eta \nabla f(\boldsymbol{w}^t)$. However, in FL, client-drift effect is a common challenge. For example, in FedAvg, the local updates $\boldsymbol{w}_{k,\ell+1}^t = \boldsymbol{w}_{k,\ell}^t - \eta \nabla f_k(\boldsymbol{w}_{k,\ell}^t)$ converge to minimizers for local objectives. Therefore, FedAvg converges to the average of these local minimizers, which is not necessarily the minimizer for the global objective function.

To address this issue, variance reduction techniques such as SAG, SAGA, FedSVRG, and SCAFFOLD can be applied to correct client drift in local updates, thereby improving convergence. In this work, we consider specifically FedSVRG and SCAFFOLD, the details of which are given below. For convenience, we denote the mini-batch (stochastic) gradient as $\nabla f_k(\boldsymbol{w}; \zeta)$, where $\zeta$ is a random subset of client $k$'s dataset.

*FedSVRG*, Federate Stochastic Variance Reduced Gradient method, mitigates client drift by introducing a gradient correction term in the local loss function. Specifically, at iteration $t$, the central server first computes and transmits the global gradient $\nabla f(\boldsymbol{w}^t)$ to each local client. The clients then perform corrected (stochastic) gradient descent (GD) steps with the difference $\nabla f(\boldsymbol{w}^t) - \nabla f_k(\boldsymbol{w}^t; \zeta)$ as the gradient correction. This enables local clients to leverage the global gradient to account for objective function heterogeneity, and FedSVRG is shown to converge linearly for smooth and strongly convex loss functions [6, 7]. However, the global gradient computation at the server requires an additional communication round.

*SCAFFOLD* [2], the Stochastic Controlled Averaging algorithm, instead introduces the server control variate $c$ and the client control variate $c_k$, which are used to estimate the update directions for the server model and each client model, respectively. The difference $c - c_k$ estimates the client drift, which is used to correct local updates as in FedSVRG. The advantage of SCAFFOLD is that it does not require the transmission of the exact global gradient $\nabla f(\boldsymbol{w}^t)$ between server and clients, reducing one round of communication in each global iteration. Here is an example of SCAFFOLD. At iterate $\boldsymbol{w}^t$, one uses the control variates as $c_k = \nabla f_k(\boldsymbol{w}^{t-1})$, $c = \nabla f(\boldsymbol{w}^{t-1})$ with information from the last iteration, update $c_k = \nabla f_k(\boldsymbol{w}^t)$, and send $c_k$ to the server along with the new local update $\boldsymbol{w}_k^t$. The server updates server variate accordingly as $c = \sum_k \frac{N_k}{N} c_k = \nabla f(\boldsymbol{w}^t)$ and $\boldsymbol{w}^{t+1} = \sum_k \frac{N_k}{N} \boldsymbol{w}_k^t$. Compared to FedSVRG, SCAFFOLD only converges sublinearly. Note that different control variates can be used. In this paper, we consider $c_k = \nabla f_k(\boldsymbol{w}^{t-1})$ and $c = \nabla f(\boldsymbol{w}^{t-1})$.

Both FedSVRG and SCAFFOLD significantly improve convergence over FedAvg [2, 7]. The summary of FedSVRG and SCAFFOLD is as follows: the local updates can be regarded as (stochastic) gradient descent on the following objective functions:

- **FedSVRG**: $f_k(\boldsymbol{w}) + \langle \nabla f(\boldsymbol{w}^t) - \nabla f_k(\boldsymbol{w}^t), \boldsymbol{w} \rangle$;
- **SCAFFOLD**: $f_k(\boldsymbol{w}) + \langle c - c_k, \boldsymbol{w} \rangle$;
- **SCAFFOLD (this paper)**: $f_k(\boldsymbol{w}) + \langle \nabla f(\boldsymbol{w}^{t-1}) - \nabla f_k(\boldsymbol{w}^{t-1}), \boldsymbol{w} \rangle$.

The gradients of these new local functions approximate the global gradients $\nabla f(\boldsymbol{w})$ such that when $\boldsymbol{w}^t$ is close to the global minimizer $\boldsymbol{w}^*$, the corrected local gradient is close to global gradient $\nabla f(\boldsymbol{w}^t)$

even though the local gradients might be much larger than it. Thus, FedAvg often requires the use of a small or diminishing local stepsize to reach the global minimizer while FedSVRG and SCAFFOLD allow "aggressive" local updates and faster convergence.

Although FedSVRG and SCAFFOLD improve the convergence of FedAvg in the data heterogeneous configuration, they are still first-order FL methods and can take many iterations to converge. This motivates us to develop acceleration methods without losing the simplicity of first-order methods.

## 2.2. Anderson Acceleration

Anderson acceleration (AA) is a scheme developed to accelerate the convergence of conventional fixed-point iterations by mixing history points. Let $\boldsymbol{w} = g(\boldsymbol{w})$ denote the fixed-point mapping and $r(\boldsymbol{w}) = g(\boldsymbol{w}) - \boldsymbol{w}$ denote the residual. We note that gradient descent steps $\boldsymbol{w}^{t+1} = \boldsymbol{w}^t - \eta \nabla h(\boldsymbol{w}^t)$ to minimize a general function $h$ can be regarded as fixed-point iterations, also known as Picard iterations, on mapping $\boldsymbol{w} \mapsto \boldsymbol{w} - \eta \nabla h(\boldsymbol{w})$ whose the residual function is $-\eta \nabla h(\boldsymbol{w})$. Additionally, with a proper local learning rate $\eta$, this mapping is contractive, as shown in [2, Lemma 6].

Now, we introduce the main idea of AA. At iteration $\boldsymbol{w}^t$, assume that we have $m + 1$ history points $\{\boldsymbol{w}^{t-i}\}_{i=0}^m$ and residuals $\{r(\boldsymbol{w}^{t-i})\}_{i=0}^m$. AA is a mixing scheme that aims to find a linear combination of history points that has minimal residual. The mixing coefficient $\boldsymbol{\alpha}$ is determined by solving a constrained least squares (LS) problem

$$\boldsymbol{\alpha}^t = \arg \min_{\boldsymbol{\alpha} \in \mathbb{R}^{m+1}} \big\| \sum_{i=0}^m \alpha_i r(\boldsymbol{w}^{t-i}) \big\|^2 \quad \text{s.t.} \quad \sum_{i=0}^m \alpha_i = 1. \tag{2}$$

The AA update $\boldsymbol{w}^{t+1}$ is computed by mixing the fixed-point evaluations of history points with coefficients $\boldsymbol{\alpha}$, i.e.,

$$\boldsymbol{w}^{t+1} = \sum_{i=0}^m \alpha_i^t g(\boldsymbol{w}^{t-i}) = \sum_{i=0}^m \alpha_i^t \boldsymbol{w}^{t-i} + \sum_{i=0}^m \alpha_i^t r(\boldsymbol{w}^{t-i}), \tag{3}$$

which is often closer to the solution than the history points or their linear combination $\sum_{i=0}^m \alpha_i^t \boldsymbol{w}^{t-i}$.

AA can be reformulated as a multisecant quasi-Newton method. Specifically, let $\boldsymbol{S} := [\boldsymbol{s}_0, \boldsymbol{s}_1, \ldots, \boldsymbol{s}_{m-1}] \in \mathbb{R}^{d \times m}$ and $\boldsymbol{Y} := [\boldsymbol{y}_0, \boldsymbol{y}_1, \ldots, \boldsymbol{y}_{m-1}] \in \mathbb{R}^{d \times m}$ with $\boldsymbol{s}_i = \boldsymbol{w}^{t-i} - \boldsymbol{w}^{t-i-1}$ and $\boldsymbol{y}_i = r(\boldsymbol{w}^{t-i}) - r(\boldsymbol{w}^{t-i-1})$. The AA update $\boldsymbol{w}^{t+1}$ in (3) can be written as

$$\boldsymbol{w}^{t+1} = \boldsymbol{w}^t - \boldsymbol{H}^{-1} r(\boldsymbol{w}^t), \tag{4}$$

$$\text{where} \quad \boldsymbol{H}^{-1} = \boldsymbol{I} + (\boldsymbol{S} - \boldsymbol{Y})(\boldsymbol{Y}^T \boldsymbol{Y})^{-1} \boldsymbol{Y}^T. \tag{5}$$

It can be verified that $\boldsymbol{H}^{-1}$ is an approximate Hessian inverse that satisfies the inverse multisecant equation $\boldsymbol{H}^{-1} \boldsymbol{Y} = \boldsymbol{S}$. Thus, AA can exploit the curvature information to accelerate convergence as other quasi-Newton methods. Meanwhile, we have $\|r(\boldsymbol{w}^t) - \boldsymbol{Y}(\boldsymbol{Y}^T \boldsymbol{Y})^{-1} \boldsymbol{Y}^T r(\boldsymbol{w}^t)\| = \|(\boldsymbol{I} - \mathrm{Proj}_{\boldsymbol{Y}}) r(\boldsymbol{w}^t)\| = \|\sum_{i=0}^m \alpha_i^t r(\boldsymbol{w}^{t-i})\|$. The convergence analysis of AA has been a hot topic in recent years. The local convergence of AA has been established in [24–27], and AA improves the convergence rate of a fixed-point iteration to first order by a factor of the *optimization gain* [26], a special term arising in AA defined as $\|(\boldsymbol{I} - \mathrm{Proj}_{\boldsymbol{Y}}) r(\boldsymbol{w}^t)\| / \|r(\boldsymbol{w}^t)\|$.

The simplicity and improved convergence rate of AA makes it an attractive choice for accelerating machine learning applications [20, 21, 28]. However, its application to distributed optimization problems, such as FL, has been relatively limited [29, 30] and remains an area ripe for further exploration. In practice, AA may become numerically unstable if performed at each iteration. Instead, a recent work, [22], shows that the alternating Anderson-Picard (AAP) method, where multiple fixed-point iterations are performed between the AA steps, is more stable in many scenarios. More interestingly, AAP is shown to approximate the Newton-GMRES update after each AA step [22, Theorem 4.5]. At the iterate $\boldsymbol{w}^t$, as an example, given $m$ history points generated by fixed-point iteration satisfying $\boldsymbol{w}^{t-i+1} = g(\boldsymbol{w}^{t-i}), i = 1, 2, \cdots m$, the AA update is equivalent to $\boldsymbol{w}^{t+1} = \boldsymbol{w}^t - \boldsymbol{p}^t - [\boldsymbol{B}^t \boldsymbol{p}^t - r(\boldsymbol{w}^t)]$, where

$$\boldsymbol{p}^t := \arg \min_{\boldsymbol{p} \in \mathcal{K}_m(\boldsymbol{B}^t, r(\boldsymbol{w}^t))} \|\boldsymbol{B}^t \boldsymbol{p} - r(\boldsymbol{w}^t)\|^2$$

is the multisecant-GMRES direction. Here, $\boldsymbol{B}^t$ is a multisecant matrix satisfying $\boldsymbol{B}^t \boldsymbol{S} = \boldsymbol{Y}$, and it converges to the Jacobian $r'(\boldsymbol{w}^t)$ when the residual $r(\boldsymbol{w}^t)$ goes to zero. The notation $\mathcal{K}_m(\boldsymbol{A}, \boldsymbol{b})$ represents the $m$-th Krylov subspace generated by the matrix $\boldsymbol{A}$ and the vector $\boldsymbol{b}$, thus $\boldsymbol{p}^t$ closely approximates the Newton-GMRES direction when the residual is small. Meanwhile, the optimization gain of the AA step converges to the Newton-GMRES gain when GMRES($m$) is employed to solve for a Newton direction at $\boldsymbol{w}^t$:

$$\frac{1}{\|r(\boldsymbol{w}^t)\|} \min_{\boldsymbol{p} \in \mathcal{K}_m(r'(\boldsymbol{w}^t), r(\boldsymbol{w}^t))} \|r'(\boldsymbol{w}^t)\boldsymbol{p} - r(\boldsymbol{w}^t)\|, \tag{6}$$

as shown in [22, Theorem 4.8]. We note that the Newton-GMRES direction here is equivalent to the Newton-MINRES method since the Hessian is symmetric [23]. This approach has been shown to be efficient and robust by introducing multiple "slow" fixed-point iterations before each AA step [22].

### 2.3. Proposed Algorithm: FedOSAA

Motivated by the success of Newton-type FL methods and favorable properties of AAP, we propose, during local updates, applying one-step AA after multiple variance-reduced first-order iterations to extrapolate new points and accelerate convergence, aptly called *Federated One-Step Anderson Acceleration* (FedOSAA).

The algorithm can be regarded as a Federated AAP method. The variance-reduced first-order iterations serve to generate local points $\boldsymbol{w}_{k,\ell}^t$ by (stochastic) gradient descent steps, while the AA step leverages curvature information from these points and approximately performs the Newton-GMRES step on the corrected local functions. This enables the algorithm to achieve convergence performance comparable to that of second-order methods such as GIANT, which follows a local Newton-CG direction on the corrected local functions; see appendix C for more details. Additionally, FedOSAA can be easily integrated into existing variance-reduced first-order FL algorithms, enhancing convergence across various local learning rates, as demonstrated in the numerical examples. However, it is important to note that FedOSAA is not applicable to FedAvg, which has no gradient correction during local updates (Appendix D.4).

We provide two examples of the proposed FedOSAA algorithm. The first is FedOSAA-SVRG, which applies AA to the FedSVRG algorithm, as presented in Algorithm 1. The second algorithm is FedOSAA-SCAFFOLD, which applies AA to the SCAFFOLD algorithm, as shown in Algorithm 2 in the appendix. The FedOSAA algorithm is highly flexible and there are options to further improve the performance; see the discussion in Appendix A.

## 3. Theoretical Analysis

In this section, we analyze the convergence properties of the proposed FedOSAA algorithm, referred to as FedOSAA-SVRG with full-batch gradients used in its gradient descent steps, i.e., $B_k = N_k$.

**Assumption 1.** *Assume that each local function $f_k$ is $\beta$-smooth and $\mu$-strongly convex, and that its Hessian $\nabla^2 f_k(\boldsymbol{w})$ is Lipschitz continuous. Additionally, the local learning rate satisfies $\eta < \frac{1}{\beta}$.*

Most existing FL algorithms provide convergence guarantees under similar assumptions. For example, FedSVRG guarantees linear convergence to the global minimum under smooth and strongly convex conditions [7]. Similarly, GIANT is shown to have local linear-quadratic convergence guarantee under smooth and strongly convex assumptions [13]. Meanwhile, the assumption that $f_k$ has a Lipschitz continuous Hessian is common in Newton-type methods. Furthermore, the upper bound on $\eta$ guarantees that the mapping $\boldsymbol{w} \mapsto \boldsymbol{w} - \eta \nabla f_k(\boldsymbol{w})$ is contractive:

$$\| (\boldsymbol{u} - \eta \nabla f_k(\boldsymbol{u})) - (\boldsymbol{v} - \eta \nabla f_k(\boldsymbol{v})) \| \leq \sqrt{1 - \eta \mu} \|\boldsymbol{u} - \boldsymbol{v}\|, \qquad \forall \boldsymbol{u}, \boldsymbol{v}, \tag{8}$$

as shown in Lemma 6 of [2]. The contractiveness of the mapping and the following assumptions on the matrices $\boldsymbol{S}_k^t$, used in the AA step during local updates, are crucial for ensuring the convergence of the AAP method; see Theorem 5.5 of [22].

**Assumption 2.** *The condition number of $\boldsymbol{S}_k^t$ is uniformly bounded for all $t$ and $k$.*

**Algorithm 1** FedOSAA-SVRG: Federated One-Step Anderson Acceleration with SVRG

---

1: **Initialization:** $\boldsymbol{w}^0 \in \mathbb{R}^d$, batchsize $B_k$, local learning rate $\eta$, local epoch $L$
2: **for** *global iteration* $t = 0$ **to** $T$ **do**
3:      /∗ Central Server                                                              ∗/
4:      Compute global gradient $\nabla f(\boldsymbol{w}^t) = \sum_{k=1}^K \frac{N_k}{N} \nabla f_k(\boldsymbol{w}^t)$
5:      Broadcast $\boldsymbol{w}^t$ and $\nabla f(\boldsymbol{w}^t)$ to local clients
6:      /∗ Local Updates in Parallel                                           ∗/
7:      **for** *each client* $k \in [1, K]$ **do**
8:          $\boldsymbol{w}_{k,0}^t \leftarrow \boldsymbol{w}^t$
9:          /∗ Take $L$ local steps with corrected gradients             ∗/
10:          **for** $\ell = 0$ **to** $L - 1$ **do**
11:              /∗ Compute the same mini-batch gradients where $|\zeta_{k,\ell}| = B_k$      ∗/
12:              $\boldsymbol{r}_{k,\ell}^t \leftarrow \nabla f_k(\boldsymbol{w}_{k,\ell}^t; \zeta_{k,\ell}) - \nabla f_k(\boldsymbol{w}^t; \zeta_{k,\ell}) + \nabla f(\boldsymbol{w}^t)$
13:              $\boldsymbol{w}_{k,\ell+1}^t \leftarrow \boldsymbol{w}_{k,\ell}^t - \eta \boldsymbol{r}_{k,\ell}^t$
14:          **end for**
15:          /∗ Take one Anderson Acceleration step                          ∗/
16:          Set $\boldsymbol{S}_k^t \leftarrow [\boldsymbol{s}_0, \boldsymbol{s}_1, \ldots, \boldsymbol{s}_{L-1}]$ with $\boldsymbol{s}_\ell \leftarrow \boldsymbol{w}_{k,\ell+1}^t - \boldsymbol{w}_{k,\ell}^t$
17:          Set $\boldsymbol{Y}_k^t \leftarrow [\boldsymbol{y}_0, \boldsymbol{y}_1, \ldots, \boldsymbol{y}_{L-1}]$ with $\boldsymbol{y}_\ell \leftarrow \boldsymbol{r}_{k,\ell+1}^t - \boldsymbol{r}_{k,\ell}^t$
18:          Compute $\boldsymbol{w}_k^t \leftarrow \boldsymbol{w}^t - \boldsymbol{H}_k^{-1} \nabla f(\boldsymbol{w}^t)$ where

$$\boldsymbol{H}_k^{-1} \leftarrow \eta \boldsymbol{I} + (\boldsymbol{S}_k^t - \eta \boldsymbol{Y}_k^t)[(\boldsymbol{Y}_k^t)^T \boldsymbol{Y}_k^t]^{-1}(\boldsymbol{Y}_k^t)^T \tag{7}$$

19:          Send local update $\boldsymbol{w}_k^t$ to the central server
20:      **end for**
21:      /∗ Central Server                                                              ∗/
22:      Update $\boldsymbol{w}^{t+1} \leftarrow \sum_k \frac{N_k}{N} \boldsymbol{w}_k^t$
23: **end for**

---

### 3.1. Convergence Analysis

In this subsection, we first show the gain from the AA step on the local updates and then provide the convergence analysis of FedOSAA. Recall that the local points are equivalent to GD iterations on the corrected local function:

$$f_k^t(\boldsymbol{w}) := f_k(\boldsymbol{w}) + \langle \nabla f(\boldsymbol{w}^t) - \nabla f_k(\boldsymbol{w}^t), \boldsymbol{w} \rangle,$$

satisfying $\nabla f_k^t(\boldsymbol{w}^t) = \nabla f(\boldsymbol{w}^t)$ and $\nabla^2 f_k^t(\boldsymbol{w}^t) = \nabla^2 f_k(\boldsymbol{w}^t)$. For the AA step, we define the local *optimization gain* as

$$\theta_k^t := \frac{\|(\boldsymbol{I} - \mathrm{Proj}_{\boldsymbol{Y}_k^t})\nabla f_k^t(\boldsymbol{w}^t)\|}{\|\nabla f_k^t(\boldsymbol{w}^t)\|} = \frac{\|(\boldsymbol{I} - \mathrm{Proj}_{\boldsymbol{Y}_k^t})\nabla f(\boldsymbol{w}^t)\|}{\|\nabla f(\boldsymbol{w}^t)\|} \le 1, \tag{9}$$

which converges to the local Newton-GMRES gain defined in (10). We further define constants

$$\delta_k^t := \frac{\|\nabla f_k^t(\boldsymbol{w}_k^t)\|}{\|\nabla f_k^t(\boldsymbol{w}^t)\|} = \frac{\|\nabla f_k^t(\boldsymbol{w}_k^t)\|}{\|\nabla f(\boldsymbol{w}^t)\|}, \quad \text{and} \quad \rho^t := \min_k \left[ \frac{(1-\delta_k^t)^2}{L} - \frac{(1+\delta_k^t)\delta_k^t}{\mu} - \frac{L(1+\delta_k^t)^2}{2\mu^2} \right].$$

**Lemma 3.** *Under Assumptions 1 and 2, given that $\|\nabla f(\boldsymbol{w}^t)\|$ is sufficiently small, we have $\delta_k^t \approx \sqrt{1-\mu\eta}\,\theta_k^t \le 1$. Further, if $f_k$ is quadratic, $\delta_k^t = \sqrt{1-\mu\eta}\,\theta_k^t \le 1$.*

The following theorems demonstrate that our algorithm converges to the global minimum $f^*$ with a linear convergence rate in two scenarios according to Lemma 3.

**Theorem 4** (Quadratic loss). *Under Assumptions 1 and 2, assume that each $f_k$ is quadratic. If $0 < \rho^t < 1$,*

$$f(\boldsymbol{w}^{t+1}) - f^* \le (1 - \frac{\rho^t}{2\mu})(f(\boldsymbol{w}^t) - f^*).$$

**Theorem 5** (General loss). *Under Assumptions 1 and 2, if $\boldsymbol{w}^0$ is sufficiently close to the global minimizer and $0 < \rho^t < 1$, we have*

$$f(\boldsymbol{w}^{t+1}) - f^* \leq (1 - \frac{\rho^t}{2\mu})(f(\boldsymbol{w}^t) - f^*).$$

The proofs of the results above can be found in Appendix B.

## 3.2. The Constant $\delta_k^t$

We can see that the convergence rate is directly influenced by the constant $\delta_k^t$, which is shaped by the product of two terms: $\sqrt{1 - \eta\mu}$ and the local optimization gain $\theta_k^t$. The first term $\sqrt{1 - \eta\mu}$ comes from the Lipschitz constant of the fixed-point mapping as in Equation (8). The local optimization gain can be estimated by the local Newton-GMRES gain. Specifically, based on Theorem 4.8 of [22], and given that $\nabla f_k^t(\boldsymbol{w}^t) = \nabla f(\boldsymbol{w}^t)$ and $\nabla^2 f_k^t(\boldsymbol{w}^t) = \nabla^2 f_k(\boldsymbol{w}^t)$, when $\nabla f_k^t(\boldsymbol{w}^t)$ is sufficiently small, $\theta_k^t$ converges to the local Newton-GMRES gain

$$\hat{\theta}_k^t := \frac{1}{\|\nabla f(\boldsymbol{w}^t)\|} \min_{p \in \mathcal{K}_L(\nabla^2 f_k(\boldsymbol{w}^t), \nabla f(\boldsymbol{w}^t))} \|\nabla^2 f_k(\boldsymbol{w}^t)p - \nabla f(\boldsymbol{w}^t)\|. \tag{10}$$

When $\nabla^2 f_k(\boldsymbol{w}^t)$ is positive definite, this Newton-GMRES gain can be bounded by $\hat{\theta}_k^t \leq 2\left(\frac{\sqrt{\kappa}-1}{\sqrt{\kappa}+1}\right)^L = 2\left(1 - \frac{2}{\sqrt{\kappa}+1}\right)^L$, where $\kappa$ is the condition number of $\nabla^2 f_k(\boldsymbol{w}^t)$ [23]. The distance between $\theta_k^t$ and $\hat{\theta}_k^t$ is bounded by $\mathcal{O}(\eta\|\nabla f_k^t(\boldsymbol{w}^t)\|)$ [22], thus we have

$$\theta_k^t \leq 2\left(\frac{\sqrt{\kappa}-1}{\sqrt{\kappa}+1}\right)^L + \mathcal{O}(\eta\|\nabla f_k^t(\boldsymbol{w}^t)\|).$$

Recall that $\nabla f_k^t(\boldsymbol{w}^t) = \nabla f(\boldsymbol{w}^t)$. Thus, when the global gradient $\nabla f(\boldsymbol{w}^t)$ is small enough, we have

$$\delta_k^t \lesssim 2\sqrt{1 - \eta\mu}\left(1 - \frac{2}{\sqrt{\kappa}+1}\right)^L.$$

A smaller learning rate $\eta$ may slightly worsen the bound and slow the convergence as shown in the numerical section. Further discussion on the comparison of FedOSAA with existing work such as DANE, GIANT, FedSVRG can be found in Appendix C, as well as the convergence properties of other FedOSAA variants.

## 4. Numerical Experiments

In this section, we show numerical comparisons on regularized logistic regression problems to show the superiority of the proposed FedOSAA algorithms. More experiments such as the performance of training neural networks (NNs) can be found in Appendix D. By default, the data are randomly and equally partitioned among $K$ local clients, mimicking an IID (independently and identically distributed) data distribution. The loss functions are still heterogeneous under this setting.

We compare the **FedOSAA** algorithms with various state-of-the-art methods, including first-order methods such as **FedAvg**, **FedSVRG** and **SCAFFOLD**, as well as second-order methods such as **L-BFGS**, **Newton-GMRES** and **GIANT**. The default FedOSAA algorithm is referred to as FedOSAA-SVRG with full-batch gradients used in its gradient descent steps. The details of these algorithms including the communication cost can be found in Appendix D.1. Here, GIANT corresponds to the Newton-CG method [13]. The larger the local epoch $L$ (the number of CG iterations, $q$, in GIANT; the number of GMRES iterations, $q$, in Newton-GMRES), the more local computation time is used. Each *aggregation round*, or equivalently, each global iteration, corresponds to two communication rounds for all algorithms mentioned above, except for SCAFFOLD and FedOSAA-SCAFFOLD. Note that both Newton-GMRES and GIANT require access to the Hessian matrix. Moreover, since the Hessian matrix is symmetric, the Newton-GMRES method is equivalent to the Newton-MINRES algorithm [31]. In all algorithms, we set the default $L = q = 10$ such that the local computation cost

of each algorithm is roughly the same, and the default learning rate $\eta = 1$. See Appendix D.3 for a detailed explanation.

The logistic regression problem with $\ell_2$ regularization can be formulated as

$$\min_{\boldsymbol{w} \in \mathbb{R}^d} \frac{1}{N} \sum_{j=1}^{N} \log \left( 1 + \exp \left( -y_j \boldsymbol{w}^T \boldsymbol{x}_j \right) \right) + \frac{\gamma}{2} \| \boldsymbol{w} \|_2^2, \tag{11}$$

where $\boldsymbol{x}_j \in \mathbb{R}^d$ is a feature vector and $y_j \in \{-1, 1\}$ is the corresponding response. For an unseen feature $\hat{\boldsymbol{x}}$, the logistic regression learns a binary classifier that makes the prediction by $\hat{y} = \mathrm{sgn} \left( \boldsymbol{w}^T \hat{\boldsymbol{x}} \right)$. This is a strongly convex problem with $\gamma$ approximating the strongly convex constant. By default, we set $\gamma = 0.001$. The FL algorithms are compared on two classical datasets: Covtype ($N = 581,012$ and $d = 54$) and w8a ($N = 49,749$ and $d = 300$). Both datasets are available at the LIBSVM website [32]. The initial point is set as $\boldsymbol{w}^0 = 0$. The results are evaluated using the relative error $\| \boldsymbol{w}^t - \boldsymbol{w}^* \| / \| \boldsymbol{w}^* \|$, where $\boldsymbol{w}^*$ denotes the global minimizer.

The first task is to evaluate the improvement of FedOSAA variants over their corresponding first-order methods, specifically FedSVRG and SCAFFOLD. As FedOSAA approximates the performance of Newton-GMRES, we also show the performance of Newton-GMRES for comparison. The evaluation is conducted by varying the local learning rate $\eta$, local epoch $L$, and the batch size $B_k$. The results, presented in Figure 1, demonstrate that FedOSAA significantly outperforms both FedSVRG and SCAFFOLD in nearly all scenarios and approximates the performance of Newton-GMRES with proper learning rates. From Figure 1 (a) and (d), we see that FedOSAA variants improve convergence across a wide range of local learning rates, even when $\eta$ is as small as 0.01. This observation aligns with our theory that FedOSAA effectively approximates the local Newton-GMRES method, even with small learning rates, thereby maintaining its efficiency without accessing Hessian, which is an advantage of FedOSAA over Newton-GMRES. Note that both methods fail to converge with a large local learning rate, e.g., $\eta = 5$, while the optimal learning rate of the first-order method is $\eta = 2/\beta \approx 4$. From Figure 1 (b) and (e), we observe that FedOSAA improves the convergence of first-order methods significantly even with only a few local epochs. Notably, FedOSAA-SVRG with $L = 3$ is comparable to FedSVRG with $L = 30$, and a similar result is seen for FedOSAA-SCAFFOLD. This shows the efficiency of FedOSAA in local computation. Finally, as shown in Figure 1 (c), FedOSAA-SVRG performs well for batch sizes $B_k$ ranging from 5 to 5810. It is important to note that there is no stochastic process when $B_k = 5810$. However, while FedOSAA-SCAFFOLD improves the convergence with full-batch gradient updates, it fails in mini-batch scenarios possibly due to inaccurate server control variate.

We also compare FedOSAA with the state-of-the-art methods under various data distributions [33]:

- **IID**: The default setting.
- **Imbalance**: In this scenario, some clients have significantly more data than others. The largest client has 50% of the data, while the smallest has only 0.2%.
- **Label-skew**: Data are randomly and almost equally partitioned, but each client has data with the same label.

In all scenarios, the local objective functions are heterogeneous. However, the latter two cases demonstrate notably higher heterogeneity, as their data distributions are significantly more diverse in our setting. Here, we set $K = 10$ for easier adjustment of the extreme imbalance ratio, although similar results are expected with $K = 100$. The result is shown in Figure 2. We can see that second-order methods consistently outperform first-order methods, regardless of the degree of heterogeneity. Both our algorithm and GIANT demonstrate comparable convergence efficiency and consistently outperform the one-step L-BFGS method, the classic quasi-Newton approach. Additionally, in the label-skewed case, GIANT may diverge, as seen in (f). However, FedOSAA still successfully finds the global minimizer with a small local learning rate and is comparable with GIANT for the first few iterations.

Further comparisons across different configurations can be found in Appendix D.

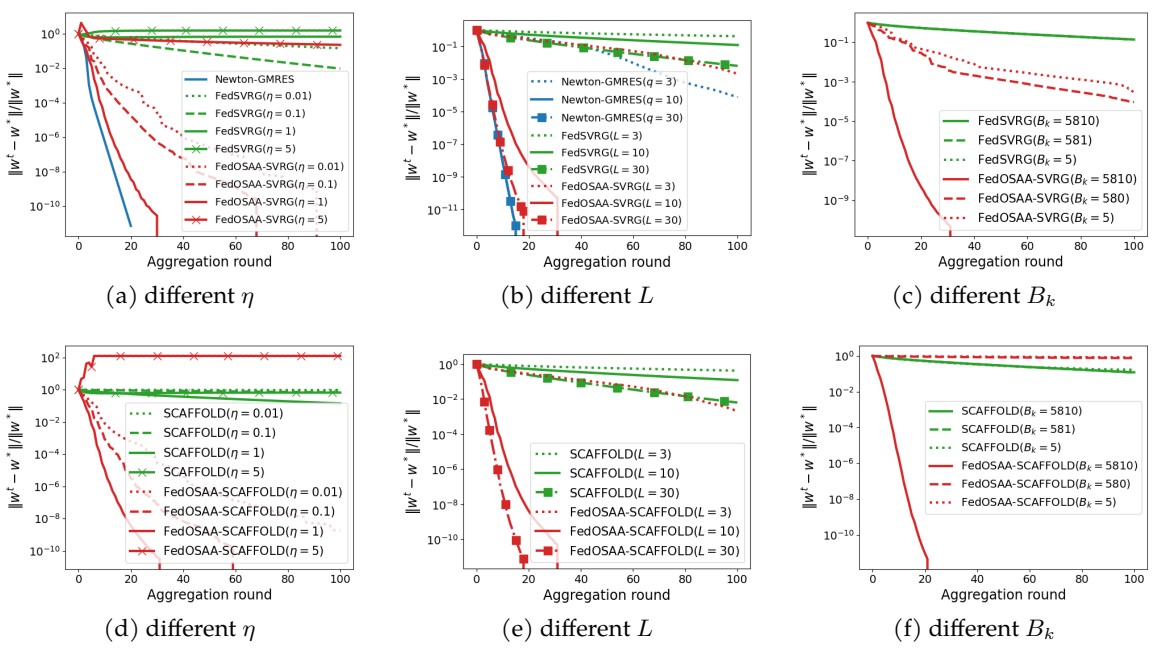

Figure 1: Comparative analysis on the `covtype` dataset by varying the local learning rate $\eta$ (first column), the number of local epochs $L$ (second column), and the batch size $B_k$ (third column). The first row compares FedOSAA-SVRG with FedSVRG and Newton-GMRES, while the second row compares FedOSAA-SCAFFOLD with SCAFFOLD. The number of clients is set to $K = 100$, with $N_k = 5810$ data points on each client.

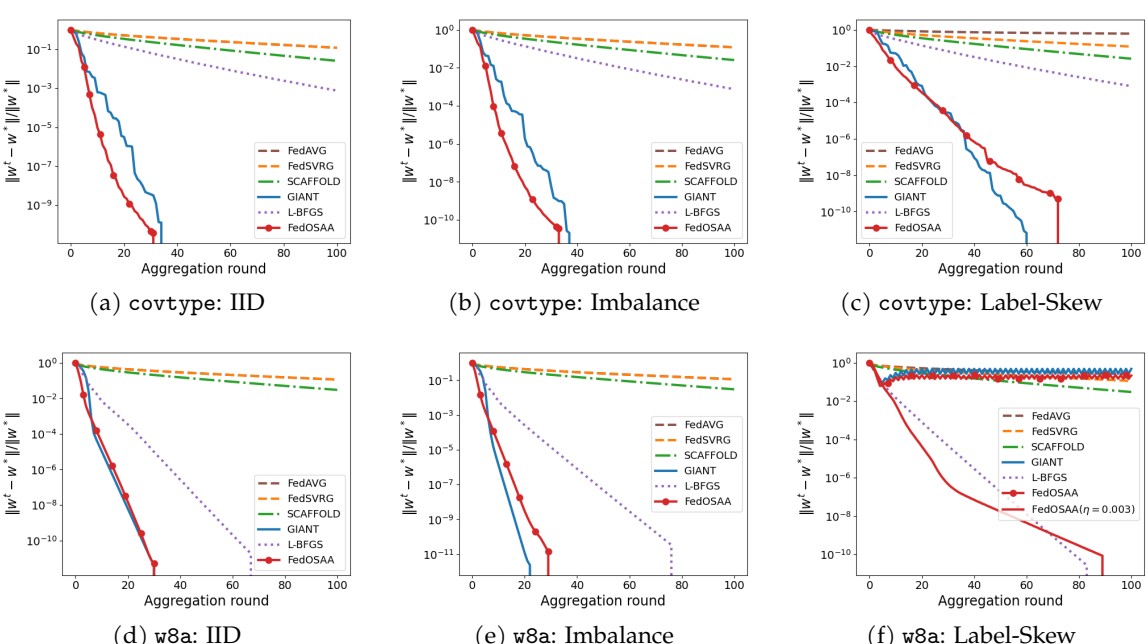

Figure 2: Comparative analysis on the different datasets and data distributions. We set $K = 10$.

# 5. Conclusion

We proposed FedOSAA, a novel scheme which accelerates the convergence of first-order FL methods by performing an AA step on each client before aggregation. The convergence property of FedOSAA is analyzed and the performance of FedOSAA is compared to other state-of-the-art FL solvers for logistic regression problems with various data distributions. The numerical results demonstrated that FedOSAA accelerates the convergence of first-order FL solvers and achieves performance comparable to a Newton-CG based solver, GIANT, without requiring Hessian information. Future work includes the incorporation of compression techniques to reduce the communication cost, investigation of privacy-preserving FL methods based on FedOSAA, and extension of FedOSAA to the partial client participation and asynchronous communication scenarios.

# Acknowledgments

This work was supported, in part, by the Office of Advanced Scientific Computing Research and performed at the Oak Ridge National Laboratory, which is managed by UT-Battelle, LLC for the US Department of Energy under Contract No. DE-AC05-00OR22725. T.S. and X.F. acknowledge support from NSF DMS-2208356, NIH R01HL16351. T.S., X.F. and P.L. acknowledge support from DE-AC05-00OR22725.

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

# A. FedOSAA-SCAFFOLD and General Strategies for Improving FedOSAA

Below is the description of FedOSAA-SCAFFOLD, where we apply one-step AA on the SCAFFOLD method [2] during local training. Here we used local client control variates $c_k = \nabla f_k(w^t)$ which is more stable.

---

**Algorithm 2** FedOSAA-SCAFFOLD: Federated One-Step Anderson Acceleration with SCAFFOLD

---

1: **Initialization:** $w^0 \in \mathbb{R}^d$, batchsize $B_k$, local learning rate $\eta$, local epoch $L$, server input $c = 0$, client $k$'s input $c_k = 0$
2: **for** *global iteration* $t = 0$ **to** $T$ **do**
3:    /$*$ Central Server                                                              $*$/
4:    Broadcast $w^t$ and $c$ to local clients
5:    /$*$ Local Updates in Parallel                                       $*$/
6:    **for** *each client* $k \in [1, K]$ **do**
7:       $w_{k,0}^t \leftarrow w^t$
8:       /$*$ *collect local points and their gradients*                          $*$/
9:       **for** $\ell = 0$ **to** $L$ **do**
10:          /$*$ Compute mini-batch gradients where $|\zeta_{k,\ell}| = B_k$            $*$/
11:          $r_{k,\ell}^t \leftarrow \nabla f_k(w_{k,\ell}^t; \zeta_{k,\ell}) - c_k + c$
12:          $w_{k,\ell+1}^t \leftarrow w_{k,\ell}^t - \eta r_{k,\ell}^t$
13:       **end for**
14:       /$*$ *take one Anderson Acceleration step*                              $*$/
15:       Set $S_k^t \leftarrow [s_0, \ldots, s_{L-1}]$ with $s_\ell \leftarrow w_{k,\ell+1}^t - w_{k,\ell}^t$
16:       Set $Y_k^t \leftarrow [y_1, \ldots, y_L]$ with $y_\ell \leftarrow r_{k,\ell+1}^t - r_{k,\ell}^t$
17:       Compute $w_k^t \leftarrow w^t - H_k^{-1} c$ where

$$H_k^{-1} \leftarrow \eta I + (S_k^t - \eta Y_k^t)[(Y_k^t)^T Y_k^t]^{-1}(Y_k^t)^T \tag{12}$$

18:       Update $c_k = \nabla f_k(w^t)$
19:       Send local update $w_k^t, c_k$ to the central server
20:    **end for**
21:    /$*$ Central Server                                                              $*$/
22:    Update $w^{t+1} \leftarrow \sum_k \frac{N_k}{N} w_k^t$, $c \leftarrow \sum_k \frac{N_k}{N} c_k$
23: **end for**

---

Below are options which may further improve the performance of FedOSAA:

- Clients can save historical points for use in the AA step in the next round.
- Use filtering techniques to remove linearly dependent columns in $Y_k^t$ and increase the stability when solving the least square problem in each AA step [34].
- Apply the moving average to the local iterations when using a stochastic gradient [28].
- Apply regularization terms in the AA LS problem, apply damping ratio in the AA step [35].
- Apply line search methods, similar to other Newton-based methods, when function evaluations are not expensive.

# B. Proofs of Section 3.1

Here we provide the proofs of Lemma 3 and Theorem 4 and 5.

**Proof of Lemma 3**.

*Proof.* Note that the local points are generated by gradient descent iterations on the corrected function $f_k^t(w) = f_k(w) + \langle \nabla f(w^t) - \nabla f_k(w^t), w \rangle$, which can be regarded as fixed-point iterations on

the mapping $g(\boldsymbol{w}) = w - \eta \nabla f_k^t(\boldsymbol{w})$. The local updates from $\boldsymbol{w}^t$ to $\boldsymbol{w}_k^t$ corresponds to one global iteration of the AAP method [22] to solve $\boldsymbol{w} = g(\boldsymbol{w})$. The residual function of the fixed-point function is $r(\boldsymbol{w}) = -\eta \nabla f_k^t(\boldsymbol{w})$. Additionally, under Assumption 1, it is a contractive fixed-point mapping with contractive constant $\sqrt{1 - \eta\mu}$.

Following Theorem 5.2 of [22], we have that when $f_k^t$ is quadratic,

$$\|\eta \nabla f_k^t(\boldsymbol{w}_k^t)\| \le \sqrt{1 - \eta\mu}\,\theta_k^t \|\eta \nabla f_k^t(\boldsymbol{w}^t)\|;$$

when $f_k^t$ is general, denoting the Hessian Lipschitz constant as $\gamma$ and the condition number of $S_k^t$ as $\kappa$, we have the residual $\|\eta \nabla f_k^t(\boldsymbol{w}_k^t)\|$ is bounded by two terms:

$$
\|\eta \nabla f_k^t(\boldsymbol{w}_k^t)\|
$$
$$
\le \sqrt{1 - \eta\mu}\,\theta_k^t \|\eta \nabla f_k^t(\boldsymbol{w}^t)\| + \sqrt{1 - (\theta_k^t)^2}\|(B_k^t)^{-1}\| \cdot \| \left[\frac{\gamma}{2}\sqrt{1 - (\theta_k^t)^2}\,\|(B_k^t)^{-1}\| + \gamma L^{1.5}\kappa\right] \|\eta \nabla f_k^t(\boldsymbol{w}^t)\|^2.
$$

Here $B_k^t$ is any multisecant matrix satisfies the multisecant equation $B_k^t S_k^t = Y_k^t$ and and there exists multisecant matrix $B_k^t$ converges to $r'(\boldsymbol{w}^t) = -\eta \nabla^2 f_k^t(\boldsymbol{w}^t)$ as the residual $\|\eta \nabla f_k^t(\boldsymbol{w}^t)\|$ goes to 0. Note that $\|r'(\boldsymbol{w})^{-1}\| \le \frac{1}{\mu}$. Thus, under Assumption 2, the coefficient of the second term is upper bounded when the residual is sufficiently small. It follows that if $\nabla f_k^t(\boldsymbol{w}^t) = \nabla f(\boldsymbol{w}^t)$ is sufficiently small, we have $\delta_k^t \approx \sqrt{1 - \mu\eta}\,\theta_k^t \le 1$.

$\square$

**Proofs of Theorem 4 and Theorem 5**

We present some useful preliminary results that will be used in the final proof:

- It is straightforward to verify that if each local function $f_k$ is $\beta$-smooth and $\mu$-strongly convex, then $f_k^t$ for all $t$ and the global function $f$ are also $\beta$-smooth and $\mu$-strongly convex. Therefore, we have the following inequalities:

$$\langle \nabla f_k^t(\boldsymbol{u}) - \nabla f_k^t(\boldsymbol{v}), \boldsymbol{u} - \boldsymbol{v} \rangle \ge \frac{1}{\beta} \left\| \nabla f_k^t(\boldsymbol{u}) - \nabla f_k^t(\boldsymbol{v}) \right\|^2, \tag{13}$$

$$\|\nabla f(\boldsymbol{w})\|^2 \ge 2\mu\left(f(\boldsymbol{w}) - f^*\right), \tag{14}$$

  for all $\boldsymbol{u}, \boldsymbol{v}, \boldsymbol{w} \in \mathbb{R}^d$. These results follow from Lemma 5 in [9].

- The Bregman divergence of a function $h$ is defined as:

$$D_h(\boldsymbol{u}, \boldsymbol{w}) = h(\boldsymbol{u}) - h(\boldsymbol{w}) - \langle \nabla h(\boldsymbol{w}), \boldsymbol{u} - \boldsymbol{w} \rangle.$$

  If $h$ is $\beta$-smooth, we have:

$$D_h(\boldsymbol{u}, \boldsymbol{w}) \le \frac{\beta}{2}\|\boldsymbol{u} - \boldsymbol{w}\|^2.$$

- Assuming $\|\nabla f_k^t(\boldsymbol{w}_k^t)\| = \delta_k^t \|\nabla f_k^t(\boldsymbol{w}^t)\|$ with $\delta_k^t \le 1$, we have
  - The gradient difference is bounded by:

$$\|\nabla f_k^t(\boldsymbol{w}_k^t) - \nabla f_k^t(\boldsymbol{w}^t)\| \ge (1 - \delta_k^t)\|\nabla f_k^t(\boldsymbol{w}^t)\|, \tag{15}$$

$$\|\nabla f_k^t(\boldsymbol{w}_k^t) - \nabla f_k^t(\boldsymbol{w}^t)\| \le (1 + \delta_k^t)\|\nabla f_k^t(\boldsymbol{w}^t)\|, \tag{16}$$

  - The points difference is bounded by:

$$\|\boldsymbol{w}_k^t - \boldsymbol{w}^t\| \le \frac{1}{\mu}\|\nabla f_k^t(\boldsymbol{w}_k^t) - \nabla f_k^t(\boldsymbol{w}^t)\| \le \frac{1 + \delta_k^t}{\mu}\|\nabla f_k^t(\boldsymbol{w}^t)\| = \frac{1 + \delta_k^t}{\mu}\|\nabla f(\boldsymbol{w}^t)\|. \tag{17}$$

*Proof.* Note that $\nabla f(\boldsymbol{w}^t) = \nabla f_k^t(\boldsymbol{w}^t)$. Assuming $\|\nabla f_k^t(\boldsymbol{w}_k^t)\| = \delta_k^t \|\nabla f(\boldsymbol{w}^t)\|$ with $\delta_k^t \leq 1$, we have,

$$
\begin{aligned}
& f(\boldsymbol{w}_k^t) \\
& = f(\boldsymbol{w}^t) + \langle \nabla f(\boldsymbol{w}^t), \boldsymbol{w}_k^t - \boldsymbol{w}^t \rangle + D_f(\boldsymbol{w}^t, \boldsymbol{w}_k^t) \\
& \leq f(\boldsymbol{w}^t) + \langle \nabla f_k^t(\boldsymbol{w}^t), \boldsymbol{w}_k^t - \boldsymbol{w}^t \rangle + \frac{\beta}{2} \|\boldsymbol{w}_k^t - \boldsymbol{w}^t\|^2 \\
& \leq f(\boldsymbol{w}^t) + \langle \nabla f_k^t(\boldsymbol{w}^t) - \nabla f_k^t(\boldsymbol{w}_k^t) + \nabla f_k^t(\boldsymbol{w}_k^t), \boldsymbol{w}_k^t - \boldsymbol{w}^t \rangle + \frac{\beta(1+\delta_k^t)^2}{2\mu^2} \|\nabla f(\boldsymbol{w}^t)\|^2 \\
& = f(\boldsymbol{w}^t) - \langle \nabla f_k^t(\boldsymbol{w}_k^t) - \nabla f_k^t(\boldsymbol{w}^t), \boldsymbol{w}_k^t - \boldsymbol{w}^t \rangle + \langle \nabla f_k^t(\boldsymbol{w}_k^t), \boldsymbol{w}_k^t - \boldsymbol{w}^t \rangle + \frac{\beta(1+\delta_k^t)^2}{2\mu^2} \|\nabla f(\boldsymbol{w}^t)\|^2.
\end{aligned}
$$

By (13), we have the second term satisfies

$$
\langle \nabla f_k^t(\boldsymbol{w}_k^t) - \nabla f_k^t(\boldsymbol{w}^t), \boldsymbol{w}_k^t - \boldsymbol{w}^t \rangle \geq \frac{1}{\beta} \|\nabla f_k^t(\boldsymbol{w}_k^t) - \nabla f_k^t(\boldsymbol{w}^t)\|^2 \geq \frac{(1-\delta_k^t)^2}{\beta} \|\nabla f(\boldsymbol{w}^t)\|^2.
$$

The third term can bounded as follows:

$$
\begin{aligned}
|\langle \nabla f_k^t(\boldsymbol{w}_k^t), \boldsymbol{w}_k^t - \boldsymbol{w}^t \rangle| & \leq \|\nabla f_k^t(\boldsymbol{w}_k^t)\| \|\boldsymbol{w}_k^t - \boldsymbol{w}^t\| \\
& \leq \delta_k^t \|\nabla f(\boldsymbol{w}^t)\| \frac{1+\delta_k^t}{\mu} \|\nabla f(\boldsymbol{w}^t)\| \\
& = \frac{\delta_k^t(1+\delta_k^t)}{\mu} \|\nabla f(\boldsymbol{w}^t)\|^2.
\end{aligned}
$$

Putting it all together, we have:

$$
f(\boldsymbol{w}_k^t) \leq f(\boldsymbol{w}^t) - \rho_k^t \|\nabla f(\boldsymbol{w}^t)\|^2 \leq f(\boldsymbol{w}^t) - \rho^t \|\nabla f(\boldsymbol{w}^t)\|^2, \tag{18}
$$

where $\rho_k^t = \left[ \frac{(1-\delta_k^t)^2}{\beta} - \frac{(1+\delta_k^t)\delta_k^t}{\mu} - \frac{\beta(1+\delta_k^t)^2}{2\mu^2} \right]$

By the convexity of the global function $f$, we have the update $\boldsymbol{w}^{t+1}$ satisfies

$$
f(\boldsymbol{w}^{t+1}) = f\left( \sum_k \frac{N_k}{N} \boldsymbol{w}_k^t \right) \leq \sum_k \frac{N_k}{N} f(\boldsymbol{w}_k^t) \leq f(\boldsymbol{w}^t) - \rho^t \|\nabla f(\boldsymbol{w}^t)\|^2 \leq f(\boldsymbol{w}^t) - \frac{\rho^t}{2\mu} \left( f(\boldsymbol{w}^t) - f^* \right),
$$

which implies:

$$
f(\boldsymbol{w}^{t+1}) - f^* \leq \left( 1 - \frac{\rho^t}{2\mu} \right) \left( f(\boldsymbol{w}^t) - f^* \right). \tag{19}
$$

According to Lemma 3, for quadratic loss, we directly have $\delta_k^t \leq 1$, and Theorem 4 follows. For general loss, given that $\|\nabla f(\boldsymbol{w}^t)\|$ is sufficiently small, we have $\delta_k^t \leq 1$ and Theorem 5 follows. $\quad\square$

# C. Related work

## C.1. Connection to Related Work

The (inexact) *DANE*, Distributed Approximate NEwton, framework [9, 12] is a well-known communication-efficient distributed method. The core idea is that each local client minimizes a surrogate local objective function at the global iteration $t$:

$$
h_k^t(\boldsymbol{w}) = f_k(\boldsymbol{w}) - \langle \nabla f_k(\boldsymbol{w}^t) - \nu_1 \nabla f(\boldsymbol{w}^t), \boldsymbol{w} \rangle + \nu_2 \|\boldsymbol{w} - \boldsymbol{w}^t\|^2.
$$

Note that $h_k^t(\boldsymbol{w})$ reduces to $f_k^t(\boldsymbol{w})$ when $\nu_1 = 1$ and $\nu_2 = 0$. If the local update $\boldsymbol{w}_k^t$ satisfies the inexactness level $\lambda \in [0, 1)$ defined as $\|\boldsymbol{w}_k^t - \hat{\boldsymbol{w}}_k^t\| \leq \lambda \|\boldsymbol{w}^{t-1} - \hat{\boldsymbol{w}}_k^t\|$, where $\hat{\boldsymbol{w}}_k^t := \arg\min h_k^t(\boldsymbol{w})$ is the minimizer of $h_k^t$, then inexact DANE converges linearly to the global minimizer with the rate:

$$
\left[ \frac{(1-\lambda)^2}{\nu_1(\beta+\nu_2)} - \frac{2\beta}{(\mu+\nu_2)^2} - \frac{2\lambda(\beta+\nu_2)}{\nu_1(\mu+\nu_2)^2} \right] \nu_1^2 \mu. \tag{20}
$$

Although this bound indicates that a smaller inexactness level $\lambda$ results in faster convergence, it is not tight due to the limitations of the proof technique. Identifying the optimal level of inexactness remains an open problem.

The local minimization problem $h_k^t(w)$ can be solved by various algorithms. Consequently, many existing methods, including our approach, fall within the scope of inexact DANE. Here, we discuss three key algorithms of interest: FedSVRG, FedOSAA-SVRG and GIANT:

- **FedSVRG**: The approximate solution is obtained by performing (stochastic) gradient descent iterations starting from $w^t$.
- **FedOSAA-SVRG**: The approximate solution is obtained by performing(stochastic) gradient descent iterations starting from $w^t$, followed by one AA step.
- **GIANT**( Globally Improved Approximate Newton Direction): The approximate solution is obtained by taking one inexact Newton step at $w^t$. Specifically, Conjugate Gradient (CG) is employed to solve the Newton problem. As a second-order approach, it requires access to the Hessian matrix.

We argue that the additional AA step in FedOSAA-SVRG enables faster achievement of the potential optimal level of inexactness compared to FedSVRG. Note that FedOSAA evaluates $L + 1$ gradients. According to Equation (8), it is straightforward to show that the local update of FedSVRG with $L+1$ local epoches satisfies:
$$\|\nabla f_k^t(w_k^t)\| \le (\sqrt{1 - \eta\mu})^{L+1} \|\nabla f_k^t(w^t)\|,$$
with the constant $(\sqrt{1 - \eta\mu})^{L+1}$ being larger than $\delta_k^t$ in FedOSAA-SVRG, as shown in Lemma 3. Considering that $f_k^t$ is strongly convex, this means that the local updates of FedOSAA-SVRG are closer to the optimal solution of $h_k^t$, leading to faster convergence. This observation is also confirmed by numerical examples.

The connection between FedOSAA-SVRG and GIANT lies in the fact that both can be viewed as local inexact Newton methods with the local update $w_k^t = w^t - p_k^t$ where
$$p_k^t \approx \arg\min_p \|\nabla^2 f_k^t(w^t)p - \nabla f_k^t(w^t)\|,$$
or equivalently, as $\nabla f_k^t(w^t) = \nabla f(w^t)$,
$$p_k^t \approx \arg\min_p \|\nabla^2 f_k(w^t)p - \nabla f(w^t)\|$$
$$= \nabla^2 f_k(w^t)^{-1}\nabla f(w^t)$$

Furthermore, as $\nabla^2 f(w^t) = \sum_k \frac{N_k}{N}\nabla^2 f_k(w^t)$, the core analysis of GIANT states that $w^{t+1} = \sum_k \frac{N_k}{N}w_k^t$ approximates an Newton step of the global function as
$$\nabla^2 f(w^t)^{-1}\nabla f(w^t) \approx \sum_k \frac{N_k}{N}\nabla^2 f_k(w^t)^{-1}\nabla f(w^t).$$

The result may also apply to FedOSAA with appropriate modifications. Furthermore, the CG method used in GIANT requires the Hessian to be positive definite, while the AA step in FedOSA can handle more general problems and does not require explicit access to the Hessian.

In summary, the one-step AA in FedOSAA improves the performance of FedSVRG and achieves a comparable performance with second-order methods.

## C.2. Further Discussion

In this section, we discuss the convergence properties of general FedOSAA algorithms.

FedOSAA-SVRG with stochastic processing can be understood as applying AA with inexact function evaluation. Specifically, at global iteration $t$, client $k$ in FedOSAA-SVRG evaluates $\nabla f_k(w_{k,\ell}^t)$. When points are generated by a stochastic process, client $k$ instead evaluates $\nabla f_k(w_{k,\ell}^t; \zeta_{k,\ell})$. Under

standard stochastic algorithm settings, we may assume $\mathbb{E}(\|\nabla f_k(\boldsymbol{w}; \zeta_{k,\ell}) - \nabla f_k(\boldsymbol{w})\|) \leq \sigma$, where $\sigma$ represents the variance of noise introduced by the stochastic process. As noted by [36], errors in function value evaluation can lead to stagnation in the convergence of AA, potentially slowing down the convergence of FedOSAA, as demonstrated in the numerical session.

FedOSAA-SCAFFOLD can be viewed as a variant of FedOSAA-SVRG where an inexact global gradient is passed to the local clients. Specifically, at global iteration $t$, FedOSAA-SVRG transmits the current global gradient $\nabla f(\boldsymbol{w}^t)$ to each client before local updates, while FedOSAA-SCAFFOLD transmits its substitute $\nabla f(\boldsymbol{w}^{t-1})$ to each client.

Both cases introduce trade-offs in terms of convergence speed and accuracy, and the analysis of other FedOSAA variants follows this.

## D. Experiments

### D.1. The Algorithms Mentioned in Section 4

Here is the summary of the algorithms.

- **FedOSAA**: There are two variants: FedOSAA-SVRG and FedOSAA-SCAFFOLD. There are three tuning parameters: the local learning rate $\eta$, the local epochs $L$, and batch size $B_k$. The default FedOSAA is FedOSAA-SVRG with no sampling, i.e, $B_k = N_k$.

- **FedAvg**: This is the benchmark FL algorithm. It has two tuning parameters: the local learning rate $\eta$, and the local epochs $L$.

- **FedSVRG**: These algorithms have two tuning parameters: the local learning rate $\eta$, and the local epochs $L$.

- **SCAFFOLD**: These algorithms have two tuning parameters: the local learning rate $\eta$, and the local epochs $L$.

- **GIANT/Newton-CG**: This algorithm has one tuning parameter: the number of CG iterations, $q$. Note that it has to access the true Hessian in a matrix-vector product mannner.

- **Newton-GMRES/Newton-MINRES**: It replace the CG solver with GMRES in GIANT. This algorithm has one tuning parameter: the number of GMRES iterations, $q$. As AAP is shown to converge to Newton-GMRES method, it servers as a reference of FedOSAA.

- **L-BFGS**: This is a special one-step L-BFGS method, where we collect local points first as in FedOSAA and then take the classical two-loop recursion of the L-BFGS method. It has two tuning parameters: the local learning rate $\eta$, and the local epochs $L$. This servers as a benchmark quasi-Newton method.

- **DANE**: It finds the exact minimizer of $f_k^t$ during local updates. We used Newton's method with line search to solve each local minimization problem. There is no tuning parameter.

The communication cost comparison is summarized in Table 1.

### D.2. Data Preparation

For the IID setting, assuming that there are $N$ data points, each client is assigned $N_k = N/K$ samples with the extra data removed.

### D.3. Computation Cost of Each Algorithm

For Logistic regression problem, the local computational cost is determined by the number of gradient evaluations. The computational cost for each full gradient evaluation is $\mathcal{O}(N_k d)$. The main computational cost of each AA step with $L+1$ history is to solve a $d \times L$ least square problem, thus $\mathcal{O}(dL^2)$. Considering $L, d \ll N_k$, the time required for the AA step is negligible. The computational

Table 1: Comparison of communication costs per aggregation round between different algorithms. Here $d$ denotes the dimension of the optimization variables (and thus the dimension of the gradients).

| Algorithm | Communication round | Communication cost |
|---|---|---|
| FedOSAA-SVRG | 2 | $2d$ |
| FedOSAA-SCAFFOLD | 1 | $2d$ |
| FedAvg | 1 | $d$ |
| FedSVRG | 2 | $2d$ |
| SCAFFOLD | 1 | $2d$ |
| GIANT | 2 | $2d$ |
| Newton-GMRES | 2 | $2d$ |
| L-BFGS | 2 | $2d$ |
| DANE | 2 | $2d$ |

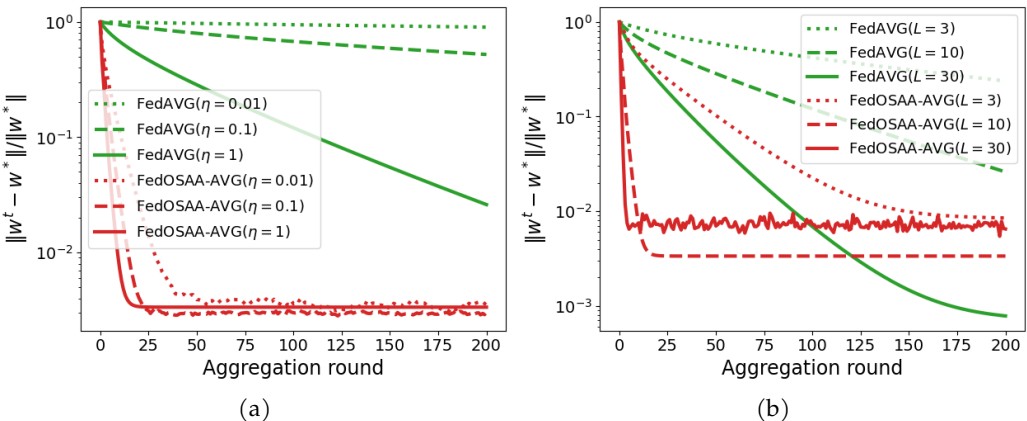

(a)           (b)

Figure 3: Comparation of FedAVG and FedOSAA-AVG on the Covtype dataset by varying the local learning rate $\eta$, the number of local epochs $L$ (second column) The number of clients is set to $K = 100$, with each client having $N_k = 5810$ data points.

cost of $L$-step CG in the GIANT method is $\mathcal{O}(LN_kd)$, which is equivalent to the computation cost of $L$ gradient evaluations. During each local training session, FedOSAA, FedAvg, FedSVRG, SCAFFOLD, and L-BFGS perform $L + 1$ gradient evaluations, whereas Newton-GMRES and GIANT require the evaluation of the Hessian and the execution of $q$ GMRES/CG iterations. Therefore, for the logistic regression problem, the local computation cost of FedOSAA, FedAvg, FedSVRG, SCAFFOLD, L-BFGS, Newton-GMRES and GIANT with the same local epochs $L$ (or $q$ in Newton-GMRES and GIANT) are roughly the same in terms of the number of FLOPs. In all algorithms, we set the default $L = q = 10$ to strike a balance between local computation and global model performance, and the default learning rate $\eta = 1$.

## D.4. Additional Experiments on Logistic Regression Problem

The first step is to demonstrate that the FedOSAA scheme is ineffective when applied to FedAVG. As is well-known, FedAVG fails to find the global minimizer. Figure 3 shows that FedOSAA-AVG fails to converge to the global minimizer under various parameter settings as well. The underlying issue is that both FedAVG and FedOSAA-AVG perform local training by minimizing the local objective functions without incorporating a gradient correction term. This makes them highly susceptible to client drift. These results highlight the necessity of incorporating a gradient correction term in local updates when applying acceleration methods to ensure convergence.

Second, we provide additional comparison FedOSAA with state-of-the-art methods on both `covtype` and `w8a` datasets in Figure 4, Figure 5 and Figure 6. It can be seen that DANE achieves

the fastest convergence in terms of the aggregation round. However, since it requires one to solve the local problem exactly, this rapid convergence comes at a high computation cost that limits its practical applicability, as shown in Figure 6. Also, DANE diverges for some examples which confirms that the optimal level of inexactness of the inexact DANE method could be greater than $0$. A poor performance of DANE is also observed in [37].

Our algorithm and GIANT demonstrate comparable convergence efficacy in terms of the aggregation round and computation time, significantly outperforming first-order methods. Furthermore, our method is constantly better than the one-step L-BFGS method, which is similar to the classic quasi-Newton method. This shows the superiority of the AA step when approximating the Hessian then other quasi-Newton methods.

Furthermore, we demonstrate the advantages of our approach on ill-conditioned problems in which a small regularization term parameter $\gamma$ gives a poorly conditioned Hessian. Typically, Newton methods, such as GIANT, require a line search to ensure convergence. We provide an example in Figure 7, where we employ the same line search strategy as described in the GIANT paper for all algorithms when indicated. We can see that GIANT with line search achieves the best performance but requires an additional round of communication and more computation time to calculate the function value. Without line search, GIANT often diverges, since the solution of Hessian inverse problem is not stable even with only $10$ CG steps. In contrast, our method without line search achieves relatively good performance.

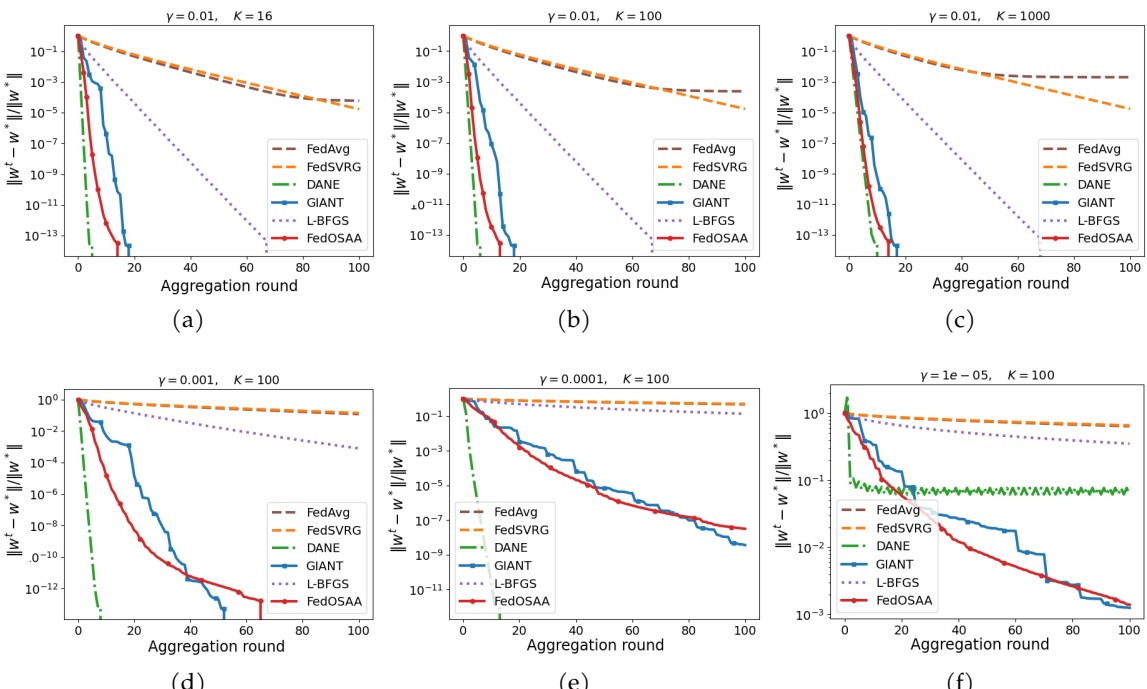

Figure 4: Comparison test on `covtype` dataset under different $\gamma$ and number of clients $K$. The first row is with a fixed $\gamma = 0.01$. The second row is with a fixed local clients $K = 100$.

## D.5. NN Training Problem

Neural network (NN) training is nonlinear, non-convex, and computationally expensive problem, and acceleration methods play a critical role in improving the efficiency of training. In this subsection, we evaluate the performance of FedOSAA on a simple NN training problem and demonstrate its tendency to get trapped in critical points. Here, FedOSAA refers to the FedOSAA-SVRG with a full-batch gradient update.

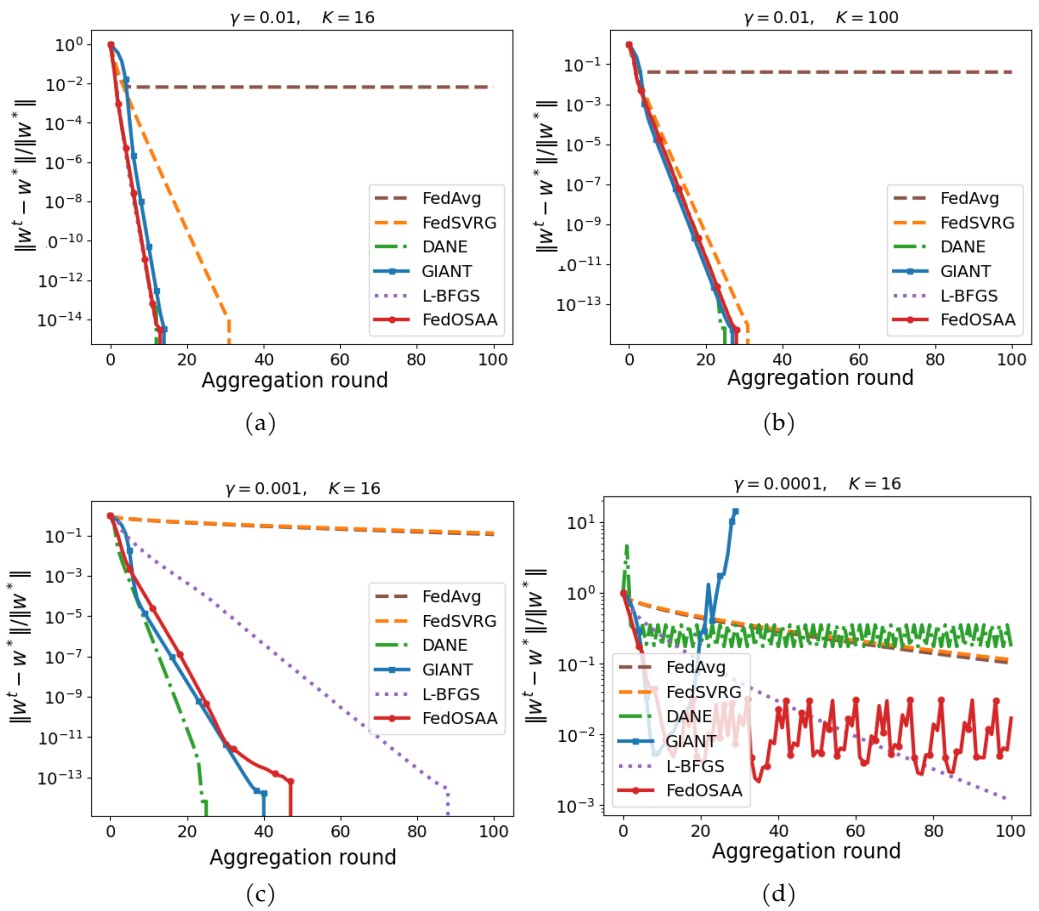

Figure 5: Comparison test on `w8a` dataset. The first row is fixed $\gamma = 0.01$. The second row is with a fixed local clients $K = 16$.

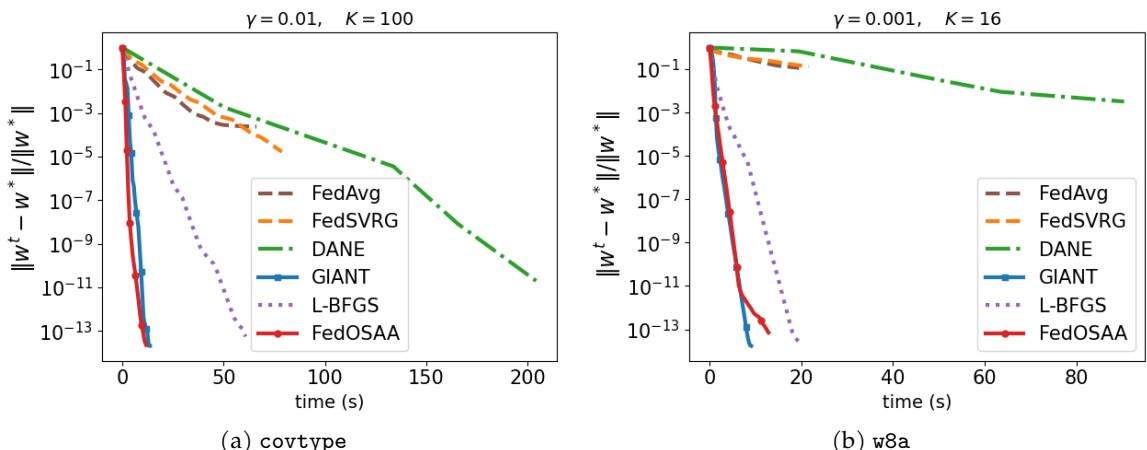

Figure 6: Comparison test in terms of computation time comparison examples. (a) is an example with `covtype` dataset. Note that the per the aggregation round cost is 51s for DANE, and around 0.8s for other algorithms. (b) is an example with `w8a` dataset. Note that the per the aggregation round cost is 20s for DANE, and around 0.2s for other algorithms.

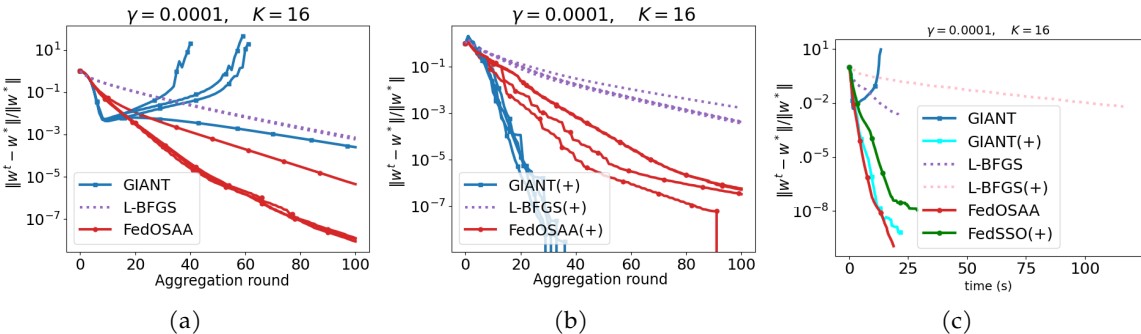

Figure 7: Comparison test on ill-conditioned problems in terms of the aggregation round and time. The result is for the `w8a` dataset and the value of regularization parameter $\gamma$ is $1e-4$. (+) means with line search in figure (c).

We consider an image classification task using the MNIST dataset. The neural network architecture is a fully connected feedforward neural network (MLP) with $N$ hidden layers, where each hidden layer consists of 256 neurons with ReLU activations. Specifically, we evaluate two cases: $N = 1$ (denoted as MLP1) and $N = 3$ (denoted as MLP3). The cross-entropy loss function is used for the classification task, and performance is assessed based on the classification accuracy on the training dataset.

We compare FedOSAA with its first-order variant, FedSVRG, and present the results in Figure 8. Under both settings, where the number of clients is $K = 1$ or $K = 10$, FedSVRG accelerates the convergence for MLP1 but fails for MLP3. A notable characteristic of FedOSAA is the rapid decrease in the global gradient norm, as shown in Figure 8(b) and (d). Recall that NN training is a non-convex problem, where the global gradient norm may remain large across iterations due to the intricate structure of the loss landscape. This behavior can be observed in Figure 8(b) and (d) where the global gradient norms of FedSVRG remain large. Especially for MLP3, the global gradient norms of FedSVRG were relatively small initially but became large later. However, the global gradient norm of FedOSAA keeps decreasing. This suggests that the iterations might be quickly attracted to a stationary point, causing the training to fail.

Second-order methods have the potential to improve convergence in non-convex problems by leveraging the curvature of the loss landscape. However, their application remains challenging. Except for the computational efficiency, existing work often employs strategies such as damping techniques or moving averages to address the non-convexity of the loss function [28]. In the context of Anderson acceleration, [35] proposed the Stochastic Anderson Mixing method, which incorporates damped projections and adaptive regularization to train various neural networks effectively. As part of our future work, we also aim to build on this approach to design an efficient yet simple algorithm for neural network training in the federated learning setting.

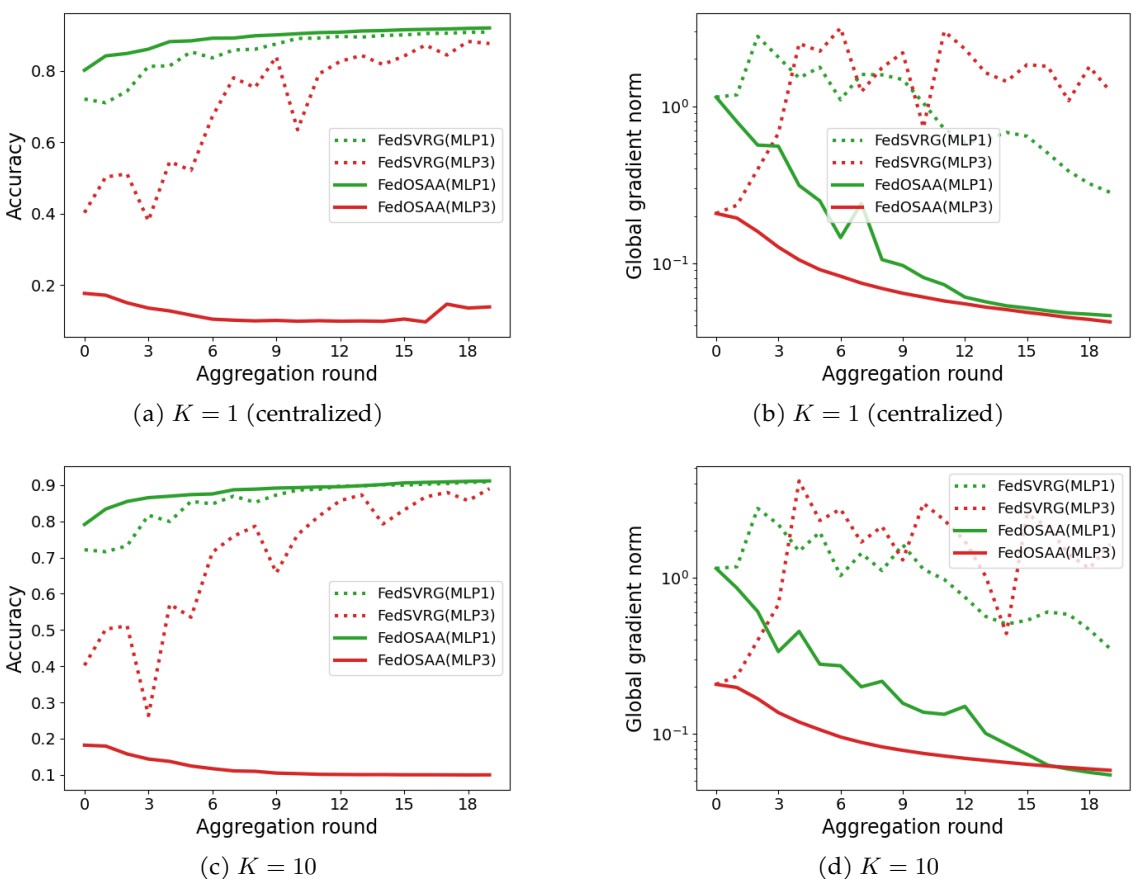

Figure 8: Comparison test on NN training problem. We set $\eta = 0.1$, $L = 10$, and $K = 1$ or $K = 10$. When $K = 1$, there is only a single local client, which mimics a centralized training setting.

