# OpenReview forum: "FedOSAA: Improving Federated Learning with One-Step Anderson Acceleration"
_CPAL.cc/2025/Proceedings_Track — CPAL 2025 (Proceedings Track) Poster_

### Official Review · Reviewer_5V2w · 2025-01-10
**New FL algorithm achieving second-order convergence and first-order computation.**

**Rating:** 6
**Confidence:** 3

**Review:**

This paper introduces FedOSAA, a novel federated learning (FL) approach inspired by Anderson acceleration. The method aims to retain the simplicity of first-order techniques while achieving the faster convergence typically associated with second-order methods. The proposed FedOSAA is evaluated against several state-of-the-art methods, including first-order approaches like FedAvg, FedSVRG, and SCAFFOLD, as well as second-order methods such as L-BFGS, Newton-GMRES, and GIANT. The methodology appears well-motivated, and the numerical analysis is thorough and convincing.

However, as I am not an expert in federated learning, there is a possibility that my review overlooks some critical evaluation criteria or nuances specific to this field. I encourage experts in FL to carefully evaluate the robustness of the claims and the broader implications of this approach.

---

### Official Review · Reviewer_rsnL · 2025-01-11

**Rating:** 6
**Confidence:** 4

**Review:**

Overview

The paper introduces FedOSAA, a novel Federated Learning (FL) algorithm that incorporates Anderson Acceleration (AA) to enhance the convergence of first-order methods like FedSVRG and SCAFFOLD. FedOSAA approximates the Newton-GMRES direction without requiring access to the Hessian matrix, achieving a convergence rate comparable to second-order methods. This improvement addresses communication and computational inefficiencies in traditional FL approaches. The paper demonstrates FedOSAA's effectiveness through theoretical analysis and numerical experiments, showing significant performance gains over state-of-the-art methods under various conditions.

Strengths

1. FedOSAA demonstrates superior performance in reducing communication rounds and computational time.

2. FedOSAA achieves rapid convergence while maintaining the simplicity of first-order methods.

Weaknesses

1. The performance under extreme label skewness shows limitations. For instance, FedOSAA-SCAFFOLD struggles with mini-batch updates in highly heterogeneous data, as noted in Fig. 2(f). This indicates sensitivity to data imbalance in practical FL settings.

2. This paper’s research is based on optimal condition. How can FedOSAA be adapted or extended to improve stability under extreme non-IID data distributions or partial client participation scenarios?

---

### Official Review · Reviewer_xYuk · 2025-01-12
**The official review of FedOSAA**

**Rating:** 7
**Confidence:** 4

**Review:**

**Summary**: This paper proposes FedSOAA,a novel federated learning algorithm that combines first-order methods with Anderson acceleration to achieve faster convergence. The key innovation is applying a one-step Anderson acceleration (AA) during local training to approximate second-order methods like Newton-GMRES, but without requiring explicit Hessian computations.

The main contributions are:

1. Integrating AA with variance reduction techniques (like FedSVRG and SCAFFOLD) to handle client drift while accelerating convergence

2. Proving local linear convergence guarantees for smooth and strongly convex loss functions

3. Demonstrating empirically that FedOSAA achieves performance comparable to second-order methods like GIANT while maintaining the simplicity of first-order methods


Prons:

- The connection between Anderson acceleration and Newton-GMRES methods is well-established
- The local linear convergence analysis is rigorous and thorough
- The approach requires no explicit Hessian computations while achieving comparable performance to second-order methods
- Clear demonstration of improved convergence rates compared to first-order methods
- Thorough ablation studies examining the impact of learning rates, local epochs, and batch sizes

Cons:

- Limited Neural Network Results
    - The neural network experiments are relatively basic (only MNIST with MLPs)
    - The method seems to struggle with deeper networks, getting trapped in critical points
    - More investigation of the limitations in non-convex settings would be valuable
- The paper could better discuss the way to address communication costs, especially with the additional storage required by historical storage

---

### Meta-Review · Area_Chair_1Ddj · 2025-02-05

**Recommendation:** Accept (Poster)
**Confidence:** 4

**Metareview:**

The paper introduces FedOSAA, a federated learning algorithm that incorporates Anderson Acceleration (AA) to improve the convergence speed of first-order methods in federated settings. The primary innovation of FedOSAA is its ability to approximate second-order optimization methods like Newton-GMRES, without the need for Hessian computations, thereby maintaining simplicity and lower computation.

It would be better if the authors could discuss more on the performance under highly heterogeneous data, and the communication costs of the method.

---

### Decision · Program_Chairs · 2025-02-11

Accept (Poster)